# Pitfalls in Monitoring Mitochondrial Temperature Using Charged Thermosensitive Fluorophores

**Dominique Chrétien [1], Paule Bénit [1], Christine Leroy [2], Riyad El-Khoury [3], Sunyou Park [4], Jung Yeol Lee [4], Young-Tae Chang [5,6], Guy Lenaers [7], Pierre Rustin [1,8] and Malgorzata Rak [1,8,\*]**

1   INSERM UMR1141, Université de Paris, NeuroDiderot, 75019 Paris, France; dominique.chretien@inserm.fr (D.C.); paule.benit@inserm.fr (P.B.); pierre.rustin@inserm.fr (P.R.)
2   INSERM UMR1151 CNRS UMR8253, Institut Necker-Enfants Malades (INEM), 75015 Paris, France; christine.leroy@inserm.fr
3   Neuromuscular Diagnostic Laboratory, Department of Pathology and Laboratory Medicine, American University of Beirut Medical Center, Beirut 11072020, Lebanon; re70@aub.edu.lb
4   New Drug Discovery Center, DGMIF, Daegu 41061, Korea; sypark6824@dgmif.re.kr (S.P.); leejysg@dgmif.re.kr (J.Y.L.)
5   Center for Self-Assembly and Complexity, Institute for Basic Science (IBS), Pohang 37673, Korea; ytchang@postech.ac.kr
6   Department of Chemistry, POSTECH, Pohang 37673, Korea
7   MitoLab Team, UMR CNRS 6015—INSERM U1083, Institut MitoVasc, Angers University and Hospital, 49933 Angers, France; guy.lenaers@inserm.fr
8   Institut des Sciences Biologiques, Centre National de la Recherche Scientifique, 75794 Paris, France
\*   Correspondence: malgorzata.rak@inserm.fr

**Abstract:** Mitochondria are the source of internal heat which influences all cellular processes. Hence, monitoring mitochondrial temperature provides a unique insight into cell physiology. Using a thermosensitive fluorescent probe MitoThermo Yellow (MTY), we have shown recently that mitochondria within human cells are maintained at close to 50 °C when active, increasing their temperature locally by about 10 °C. Initially reported in the HEK293 cell line, we confirmed this finding in the HeLa cell line. Delving deeper, using MTY and MTX (MitoThermo X), a modified version of MTY, we unraveled some caveats related to the nature of these charged fluorophores. While enabling the assessment of mitochondrial temperature in HEK and HeLa cell lines, the reactivity of MTY to membrane potential variations in human primary skin fibroblasts precluded local temperature monitoring in these cells. Chemical modification of MTY into MTX did not result in a temperature probe unresponsive to membrane potential variations that could be universally used in any cell type to determine mitochondrial temperature. Thus, the cell-type dependence of MTY in measuring mitochondrial temperature, which is likely due to the variable binding of this dye to specific internal mitochondrial components, should imply cautiousness while using these nanothermometers for mitochondrial temperature analysis.

**Keywords:** mitochondria; temperature; thermosensitive fluorescent probes; MitoThermo Yellow

## 1. Introduction

Mitochondria are cellular organelles specialized in converting free energy from oxidation of nutrients into high-energy adenosine triphosphate (ATP) and heat [1,2]. The metabolic heat, which represents most of the energy extracted from respiratory substrates, is used by endothermic species to warm up their body and to keep a stable body temperature independently of prevailing

external conditions [3,4]. The thermogenic activity of mitochondria also affects the intracellular temperature homeostasis [5–7]. It was shown recently that actively operating mitochondria within human cells increase their temperature locally by about 10 °C compared to external cell medium and are maintained at close to 50 °C [8]. Notably, this high temperature corresponds to the temperature range of maximal activity of respiratory chain complexes [8].

Temperature plays a vital role at all levels of cellular organization and function, from dictating rates of biochemical reactions to influencing physical properties of biological membranes [9,10]. Therefore, the thermogenic activity of mitochondria leading to local temperature elevation [8] is likely affecting thermal mitochondrial environment could emerge as a new major mitochondrial function.

As variations in mitochondrial thermogenesis may reflect the bioenergetics status of mitochondria [3,11–13], monitoring the temperature of mitochondria should provide a unique insight into mitochondrial functioning under various physiological and pathological conditions. At present, accurate real-time monitoring of mitochondrial temperature in living cells remains challenging (reviewed in [14–16]). Among the few molecular tools available, nano-sized temperature-sensitive fluorescent dyes represent a valuable option [7,17–19]. These lipophilic cations, derived from rhodamine, preferentially localize into mitochondria, driven by the negative charges on the matrix face of the inner mitochondrial membrane. In principle, once in the mitochondria, they can be used to measure mitochondrial temperature in any cell type and under different experimental conditions. However, as highlighted in this work, using the previously reported probe MitoThermo Yellow (MTY) and a recently developed MitoThermo X (MTX), we identified some caveats related to the nature of these charged fluorophores that restrict their use to specific cell types and imply cautiousness while studying mitochondrial temperature with these nanothermometers.

## 2. Material and Methods

### 2.1. Cell Culture

Commercially available human cell lines derived from embryonic kidney (neural lineage origin), HEK293 cells, or from cervical cancer, HeLa cells, as well as primary skin fibroblasts derived from healthy individuals were cultured in DMEM medium (Dulbecco/Vogt modified Eagle's minimal essential medium) containing 4.5 g/L glucose and 2 mM glutamine (DMEM Glutamax; Gibco Thermo Fisher Scientific, Waltham, MA, USA), 10% fetal calf serum, 200 μM uridine, 2 mM pyruvate, penicillin and streptomycin, 100 U/mL each (all from Sigma Aldrich, Saint-Quentin Fallavier, France). Cells were grown at 80–85% humidity, in a 5% $CO_2$ incubator at 37 °C. Human primary fibroblasts were derived from skin biopsies obtained from anonymous healthy donors with informed consent for research.

### 2.2. Fluorescent Probes

MitoTracker Green (MTG; Invitrogen M7514), MitoTracker Red (MTR; Invitrogen 7512) and tetramethylrhodamine, methyl ester, perchlorate (TMRM; Invitrogen T668) are commercially available. MitoThermo Yellow (MTY) was prepared as previously reported [16,20].

Synthetic Procedure for MTX

Known compound 1 (Figure S1 in the Supplementary Materials) was prepared from 4-nitrosalicylic acid according to a reported procedure [21]. Compound 1 (500 mg, 0.5 mmol) was suspended in freshly dried tetrahydrofuran (THF) (5 mL) in a sealed tube. (3-(piperidin-1-ylmethyl)phenyl)magnesium bromide (1 M in tetrahydrofuran (THF), 5 mL, 5 mmol) was added and heated at 68 °C on a heat-block for 38 h. The resin was filtered through a 5 mL cartridge and washed with dichloromethane (DCM) (X5), dimethylformamide (DMF) (X5) and methanol (MeOH) (X5). The resin was dried and treated with 5% trifluoroacetic acid (TFA) in dichloromethane/methanol (DCM/MeOH) (15:1, 15 mL) for 30 min. The solution was drained to the round bottom flask, then washed with brine (100 mL), dried over

anhydrous sodium sulfate, filtered, and concentrated in vacuo. The crude compound 2 obtained in the first step was used directly for the next step without further purification.

To a solution of compound 2 (21.1 mg, 0.048 mmol) in MeCN/DCM (1:5, 2.5 mL) was added chloroacetic anhydride (8.2 mg, 0.048 mmol) in several portions at 0 °C. The reaction mixture was stirred for 10 min, and then pyridine (15.2 μL, 0.096 mmol) was added. The reaction mixture was stirred at for 5 min at 0 °C and then quenched with aqueous solution of sodium bicarbonate (1 mL). The aqueous layer was extracted with 3% MeOH in DCM and combined organic layers were washed with brine, dried over anhydrous sodium sulfate, filtered, and concentrated in vacuo. The residue was purified by prep-HPLC (MeCN:$H_2O$, 3:97 to 75:25, 60 min) to yield MTX (10.6 mg, 0.019 mmol) as a red solid.

### 2.3. Staining Procedures and Live Cell Imaging

Cells were grown in 12-well plates for 48 h in DMEM medium, as described above. To label the cells, the culture medium was replaced by a pre-warmed medium containing extemporarily added fluorescent dyes at a final concentration of 100 nM MTG and 100 nM MTY or 100 nM MTX. After 15 min incubation in the cell culture incubator, the staining medium was replaced by fresh pre-warmed medium and cells were observed with a confocal scanning laser microscope (Leica TCS SP8; Leica Microsystems, Wetzlar, Germany) equipped with 488 nm Ar and 561 nm DPSS (Diode-Pumped Solid-State) lasers and with a temperature and $CO_2$ controller (The Brick, The Cube and The Box, Life Imaging Services, Basel, Switzerland). Eight-bit digital images were collected from a single optical plane using a HC PL APO CS2 40x/1.30 oil-immersion Leica objective. Double-fluorescence images were acquired in sequential mode to avoid fluorescence emission cross-talk.

### 2.4. Simultaneous Fluorescence, Temperature, Oxygen Uptake Assay

Detached subconfluent HEK293 cells (25 $cm^2$ flask) or trypsinized subconfluent HeLa cells or primary skin fibroblasts (75 $cm^2$ flask) were treated for 15 min with either 100 nM MTY, 100 nM MTX, 100 nM MTR or 200 mM TMRM in 10 mL DMEM and recovered by centrifugation at 470× *g* for 5 min. The pellet was washed once in 1 mL PBS 1X, then maintained as a concentrated pellet for 30 min to establish anaerobiosis. The cells (1 mg prot) were then injected into 600 or 800 μL air-saturated PBS 1X (closed or open cuvette, respectively) thermostatically maintained at 37 °C. The fluorescence (excitation 542 nm, emission 564 nm for MTY, MTX; excitation 569 nm, emission 594 nm for MTR; excitation 554 nm, emission 577 nm for TMRM), the temperature of the medium in the cuvette, and the respiration of the intact cell suspension were simultaneously measured in a 37 °C-thermostated 1 mL-quartz cell, magnetically stirred at 350 rpm (600 μL PBS, closed cuvette) or 500 rpm (800 μL PBS, open cuvette), using a modified-Xenius XC spectrofluorometer (SAFAS, Monaco). Oxygen uptake was measured with an optode device (Pyroscience, France) fitted to a 3D printed cap, ensuring either closure of the quartz cell yet allowing micro-injections (hole with 0.8-mm diameter), or leaving the quartz cell open to allow for constant oxygen replenishment. To modulate or block cell respiration either 5 μM oligomycin or 0.8 mM cyanide was added. As the probe thermosensitivity may be affected by the mitochondrial environment, the temperature differentials were quantified by a controlled temperature shift of the cell suspension medium with the water bath at the end of each experiment. By simultaneously monitoring cell respiration, the lack of interference of the probe (at least up to 300 nM) with mitochondrial function was verified during each assay. The 3D cap models were created using free Tinkercad online software and the caps were printed using Ultimaker2+ 3D printer with methylmethacrylate acrylonitrile butadiene styrene (Clearscent ABS). Of note, some ABS polymers can bind small amounts of dye and hydrophobic inhibitors. Thorough washing of the cap is thus required.

## 3. Results

In this work, we studied: HEK293 cell line (HEK), used in the original study [8], HeLa cells, another human cell line, as well as human primary skin fibroblasts, to further investigate mitochondrial temperature in intact cells.

We first verified the mitochondrial targeting and retention of MTY, as these parameters may vary among cell types and an uncontrollable leakage of the dye from mitochondria would preclude temperature analysis in these cells, as previously observed in two human cell lines [8]. We thus confirmed that MTY accumulated efficiently in the mitochondria of HEK cells and primary human fibroblasts where it was retained for at least 2 h [8] (Figure S2 in the Supplementary Materials). Of note, at the concentration of 100 nM used in this study, MTY is not toxic which remains true up to 4 μM, even after 24 h [8,18].

### 3.1. Fully Energized Mitochondria in HeLa and HEK Cells Increase Their Temperature by about 10 °C

MTY is a thermosensitive positively charged dye accumulating in the mitochondria due to the negatively charged matrix side of the inner membrane. The majority of charged fluorophores, including rhodamine and its derivatives, are sensitive to variations of the membrane potential, although to different extents. An increase in the membrane potential leads to an increase in the dye sequestration and subsequent fluorescence quenching that correlates with the degree of dye stacking. Conversely, a decrease in the membrane potential results in the dissociation of the dye into free forms and the fluorescence de-quenching [22]. Accordingly, MTY accumulates in the mitochondria attracted by the negative mitochondrial membrane potential (MMP). However, after its initial membrane potential-dependent uptake, MTY binds to its targets within the mitochondria, including the mitochondrial ALDH2 [18,23] resulting in retention of the dye in the mitochondria and consequently making it insensitive to mitochondrial potential variations. Thus the fluorescent signal of MTY is unaffected by the confounding effects of fluorescence quenching and is therefore solely influenced by changes in the surrounding temperature.

Accordingly, when HEK cells stained with MTY are injected into an oxygenated medium to activate respiration, the fluorescent signal of MTY decreases very progressively, witnessing an increase in mitochondrial temperature resulting from mitochondrial thermogenesis (Figure 1a,b). The addition of potassium cyanide (KCN), a Complex IV (CIV) inhibitor, that in a few seconds fully blocks the electron flow through the respiratory chain, leads to a progressive increase of MTY fluorescence indicating a progressively decreasing temperature. Inhibiting the respiration of the HEK cells with the ATPase synthase inhibitor, oligomycin, results in a progressive increase in fluorescence, as observed with KCN. This occurs despite the fact that oligomycin causes a hyperpolarization of the inner membrane, an opposite effect to KCN known to collapse the membrane potential. Under such conditions, an instantaneous decrease in fluorescence due to the fluorescence quenching should occur for any charged dye, sensitive to the membrane potential variations. In order to verify this behavior in HEK cells, we stained the cells with the membrane potential-sensitive probe Mitotracker Red (MTR) (Figure 1a) [24]. Accordingly, an immediate large fluorescence decrease after oligomycin addition is observed with rapid kinetics similar to those observed with probes sensitive to mitochondrial membrane potential fluctuations. Similar results are obtained with TMRM-loaded HEK cells although the decrease of oligomycin-induced fluorescence is somewhat lower and the signal is unstable when the whole cells are labeled. However when digitonin-permeabilized cells are used, after energizing mitochondria with succinate and ADP (Adenosine diphosphate), ATP synthase blockage with oligomycin results in a large and stable fluorescence decrease (Figure 1c).

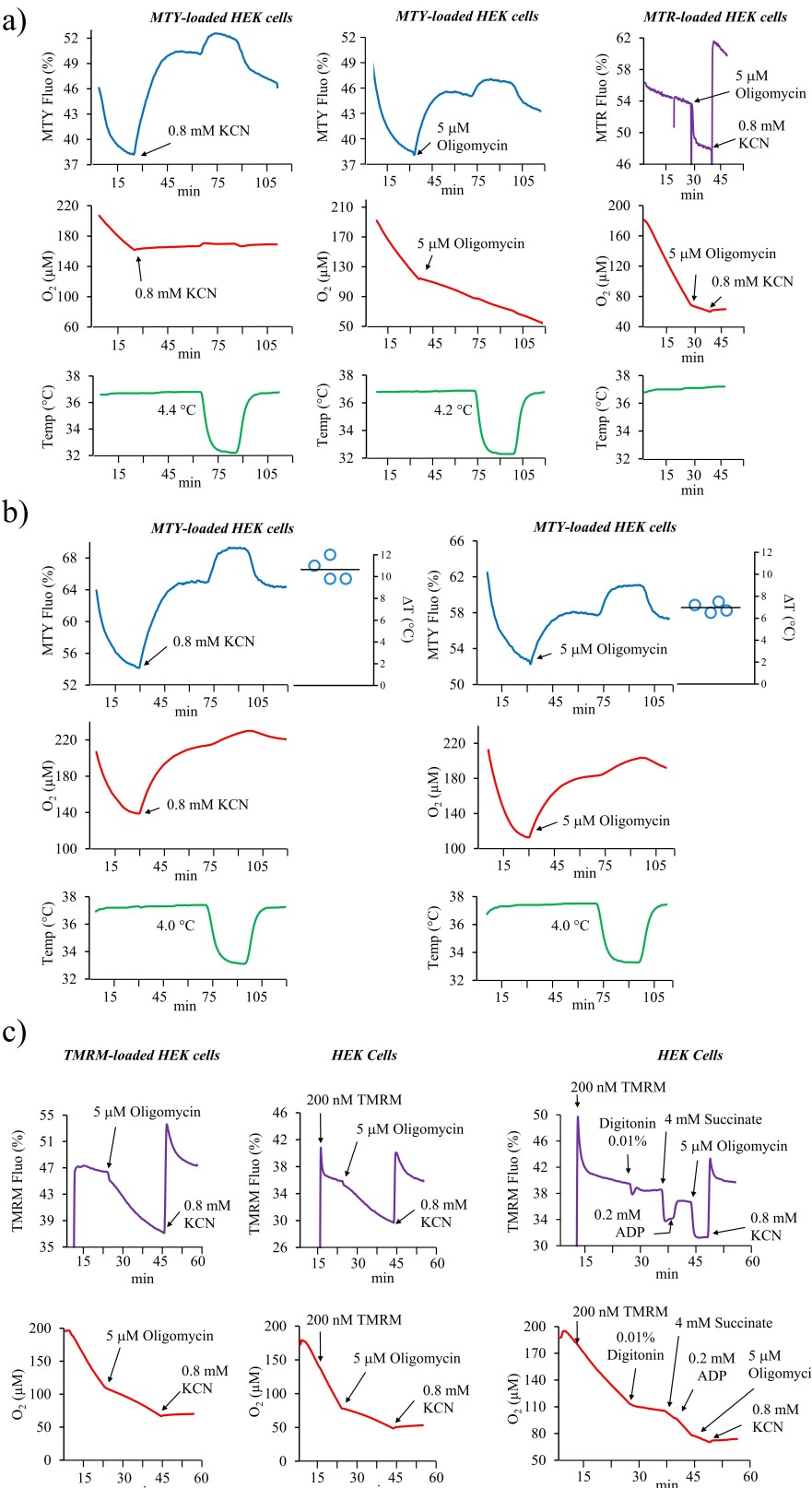

**Figure 1.** The analysis of the mitochondrial temperature or mitochondrial membrane potential in MitoThermo Yellow (MTY), MitoTracker Red (MTR), or tetramethylrhodamine, methyl ester, perchlorate (TMRM)-labeled HEK293 cells. (**a**) Activation of the respiration in MTY-labeled HEK cells causes a

progressive decrease in MTY fluorescence (blue trace) corresponding to an increasing mitochondrial temperature, whereas an inhibition of the respiration by either KCN or oligomycin results in a progressive increase in MTY fluorescence resulting from the arrest of thermogenesis and decreased mitochondrial temperature. Injection of the MTR-labeled HEK cells results in a stable fluorescent signal (violet trace) which immediately decreases after the addition of oligomycin and increases after the subsequent addition of potassium cyanide (KCN), visualizing the variation of mitochondrial membrane potential. The cell respiration (red trace) and the temperature of the cell suspension medium (green trace) are simultaneously monitored. Measurements are carried out in a closed quartz chamber and repeated 3 times (MTR) or 4 times (MTY); representative traces are shown. (**b**) The measurement of the mitochondrial temperature in MTY-labeled HEK293 cells performed as in 1a and carried out in an open quartz cuvette. The right panel represents computed temperature differential from 4 independent measurements plotted on the graph. (**c**) TMRM-loaded HEK293 cells are injected into oxygenated PBS in the spectrofluorometer cuvette (left panel) or TMRM is added directly into cuvette containing the unlabeled cell (middle panel). The cell respiration (red line) and the TMRM fluorescence (violet line) are simultaneously monitored. Alternatively, HEK cells are labeled in the respiratory buffer (right panel) then permeabilized with digitonin and subsequently energized with succinate. ATP synthase is stimulated by ADP addition followed by its inhibition with oligomycin leading to a fluorescence decrease because of mitochondrial membrane potential (MMP) hyperpolarization. Measurements are carried out in a closed quartz chamber. Representative graphs are shown. Experiments on intact or permeabilized HEK cells were performed 3 times.

Taken together, using several well-recognized mitochondrial membrane potential probes, MMP variation can be consistently detected both in permeabilized cells using R123 [8] and intact cells using MTR and TMRM (this study). Therefore the opposite behavior of MTY shows its non-responsiveness to membrane potential after accumulation into mitochondria of HEK cells and indicates an arrest of mitochondrial thermogenesis caused by the inhibition of respiration. This conclusion is further supported by the respective kinetics of MTR and TMRM fluorescence variations occurring in the range of seconds while that of MTY fluorescence occurs in the range of minutes.

Of note, MTY was used at a 100 nM concentration throughout this study, which is within typical concentration ranges of various MMP probes such as R123, from which it is derived. As our unique set up allows for simultaneous measurement of mitochondrial respiration, coupling efficiency, and sensibility to various mitochondrial inhibitors, the impact of any molecule on mitochondrial function is controlled in each of the experiments. Thus, we could verify and confirm that least up to 300 nM, MTY does not interfere with mitochondrial function.

To control the effect of inhibitors on cellular respiration, these experiments were carried out in a hermetically capped chamber (as described in [8,25]). However, for the quantitative analyses of the effect of respiratory chain inhibitors, we found it necessary to use an uncapped chamber, allowing constant oxygen replenishment and avoiding any confounding effect related to decreasing oxygen tension (Figure 1b). This was especially important when analyzing the effect of oligomycin on thermogenesis as too low an oxygen tension could lead to overestimating the temperature differential. We thus confirmed that following the activation of respiration in HEK cells, the rise of the mitochondrial temperature is about 10 °C, compared to the temperature of the surrounding cell-suspension medium maintained at 37 °C (Figure 1b). As there is a residual respiration after ATP synthase blockage, the temperature differential computed from the MTY fluorescence variation after oligomycin treatment is about 70% of that induced by the total oxygen consumption blockage triggered by KCN.

Thus, an efficient uptake of MTY by HEK cells and the subsequent binding of the dye within the mitochondria allow for a robust determination of mitochondrial temperature in these cells. Aiming to extend the mitochondrial temperature measurement to another cell type, we used HeLa cells which also showed an efficient and long term MTY staining of the mitochondrial network. Of note, after 1 h we also observed some fluorescent aggregates in the cytosol (Figure 2a). Using the same experimental approach as for HEK cells, we preloaded HeLa cells with 100 nM MTY and monitored the fluorescence

changes caused by activation or inhibition of mitochondrial respiration. Inhibition of the respiration with either KCN or oligomycin led to the same slow increase in MTY fluorescence (Figure 2b). This takes place independently of membrane potential variations and demonstrates a decreasing mitochondrial temperature resulting from arrested mitochondrial thermogenesis. The quantification of the rise in the mitochondrial temperature estimated to be around 10 °C (ranging from 9 to 11 °C) was similar to the one we measured in HEK cells (Figure 2b).

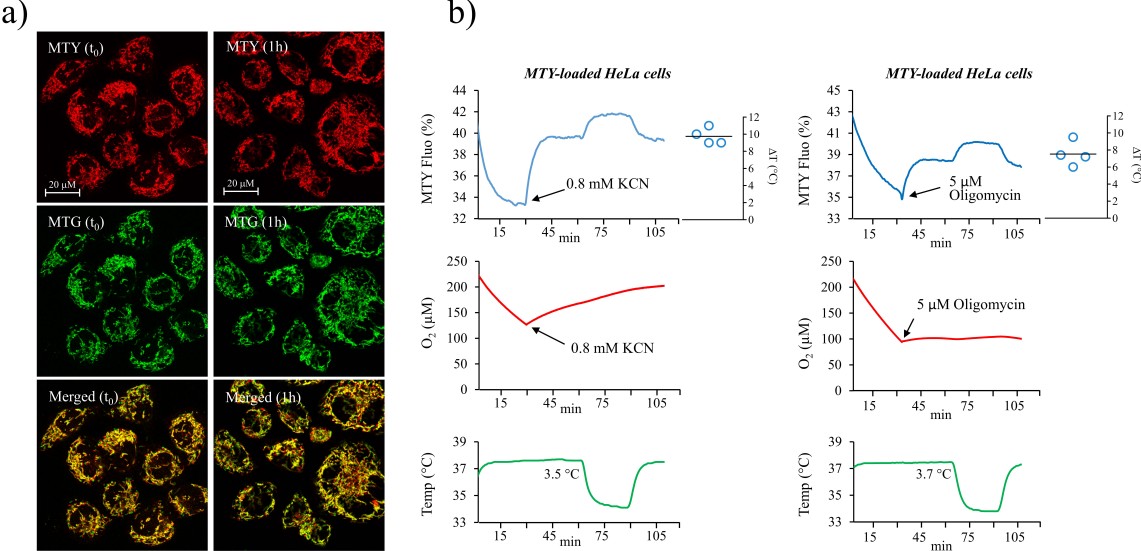

**Figure 2.** Determination of mitochondrial temperature in HeLa cells. (**a**) The MTY temperature-sensitive probe (**red**) colocalizes with MitoTracker Green (MTG; **green**) as visualized immediately, at $t_0$ or 1 h after the labeling; (**b**) analysis of the mitochondrial temperature in MTY-labeled HeLa cells (as in Figure 1b). Activation of the respiration causes a decrease in MTY fluorescence (**blue** line) corresponding to an increasing mitochondrial temperature, whereas a blockage of the respiration by either KCN or oligomycin results in a progressive increase in MTY fluorescence, resulting from an arrest of thermogenesis and a decrease of mitochondrial temperature. The respiration (**red** line) and the cell suspension temperature (**green** line) are simultaneously monitored. Measurements are carried out in an open quartz cuvette. The right panel represents computed temperature differential from 4 individual independent measurements plotted on the graph.

### 3.2. Sensitivity of MTY to Membrane Potential Variations in Human Primary Fibroblasts Prevents Its Use for Temperature Measurement in These Cells

Despite an efficient staining and a long retention of MTY in the mitochondria of primary human fibroblasts (Figure S2), its fluorescence was found to be sensitive to membrane potential variations. This was shown by the rapid fluorescence decrease after oligomycin addition which is compatible with mitochondrial hyperpolarization induced by the ATP synthase inhibition (Figure 3, middle and right panels). A subsequent addition of KCN leading to an arrest of respiration and a membrane potential collapse caused an increase in MTY fluorescence with kinetics in the range of seconds, compatible with known rates of membrane potential fluctuations. When the respiration was solely blocked with KCN (Figure 3, left panel), the same rapid MTY fluorescence increase was observed. Although the probe remained thermosensitive and responded to the subsequent calibration with the water bath, the temperature could not be quantified, as the part of the observed fluorescence signal specifically corresponding to the temperature variation is impossible to determine. Thus, in human fibroblast, the MTY sensitivity to the membrane potential variations precludes its use in monitoring mitochondrial temperature.

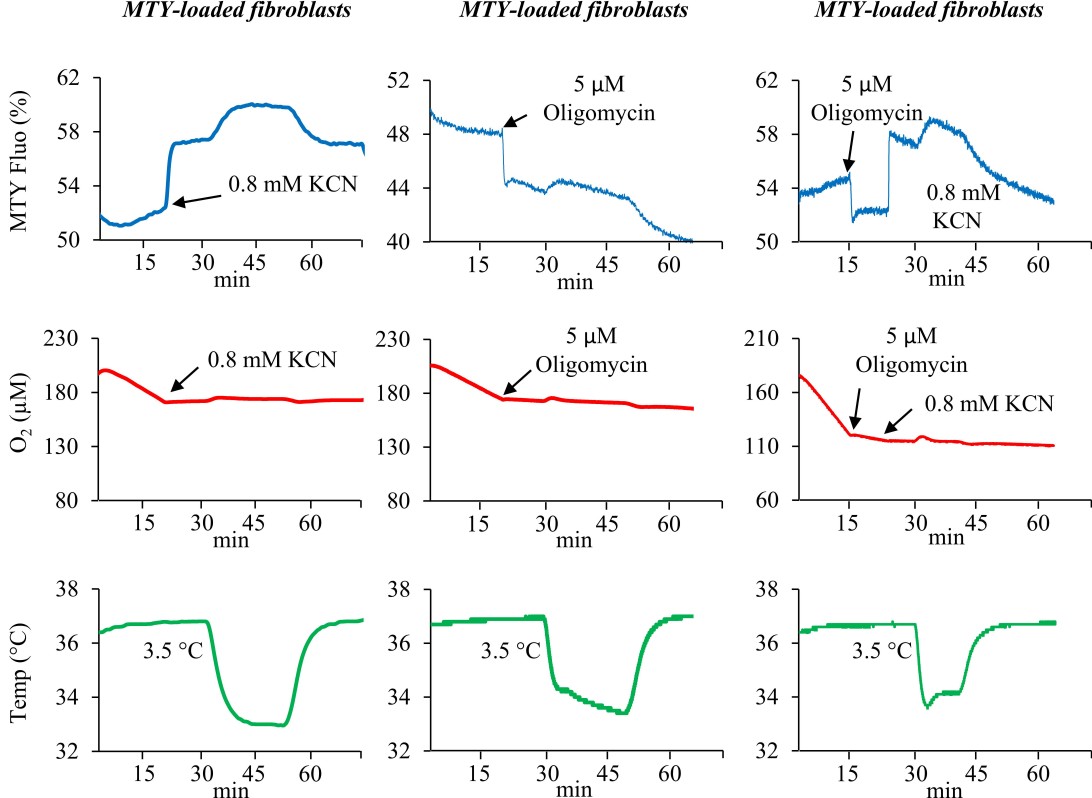

**Figure 3.** MTY and mitochondrial temperature in human primary fibroblasts. Fluorescence changes in the MTY-labeled human primary fibroblasts. Addition of the MTY-labeled fibroblasts results in a slight decrease in the fluorescence (blue lines). The inhibition of respiration by KCN results in a rapid increase in MTY fluorescence (blue line, left panel) or in a further decrease in fluorescence when oligomycin is added (blue lines, middle and right panels). Subsequent addition of KCN causes an immediate increase of the fluorescence (blue line, right panel). The respiration (red lines) and the temperature of the cell suspension (green lines) are simultaneously monitored. Measurements are carried out in a closed quartz chamber. Representative graphs from 3 independent experiments (+ KCN or + oligomycin) are shown.

### 3.3. Modification of MTY into MTX Did not Generate a Universal Temperature Probe Insensitive to Membrane Potential Variations

In an attempt to increase the stability of MTY and its anchoring into mitochondria, the probe was chemically modified by adding a chloroacetyl motif giving rise to the MTX compound (Figure 4a,d; Figure S1). The chloroacetyl motif is known to react with thiols [26], and is expected to covalently bind to the cysteine residues of nearby proteins. For instance, in a previous report, we demonstrated through mass spectrometry analysis that when added to the tubulyzine for affinity study, the chloroacetyl group could covalently bind to 12Cys(beta) of tubulin [27]. We thus hypothesized that if such covalent binding of MTX to mitochondrial proteins occurs, it should stabilize the dye and increase its retention in the mitochondria.

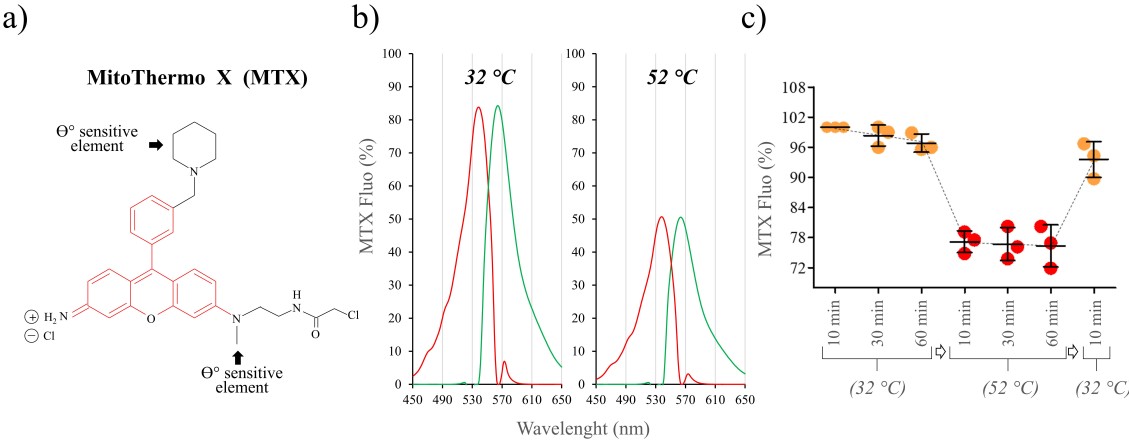

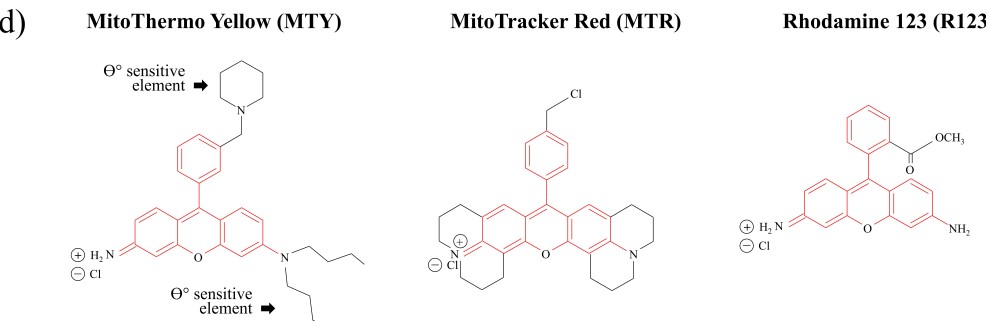

**Figure 4.** Spectral properties of the MTX probe. (**a**) Structure of MTX; (**b**) fluorescence excitation (red) and emission (green) spectra of MTX (1.5 μM) $H_2O$ at 32 °C and 52 °C; (**c**) fluorescence response of MTX in solution to temperature (% of fluorescence at 32 °C). The probe was incubated for 10, 30, and 60 min at 32 °C followed by 10, 30, 60 min incubation at 52 °C, and the temperature was decreased back to 32 °C, as indicated. n = 3; values are means ± SD; (**d**) structures of rhodamine-derived MitoThermo Yellow (MTY), MitoTracker Red (MTR) and rhodamine 123 (R123) with a common rhodamine element (red) and added motifs (black), as indicated.

The spectral characteristics of MTX are close to that of MTY, with wavelengths of maximum absorbance and emission of 536 and 563 nm at 32 °C and 52 °C (Figure 4b). Prolonged exposure to a high temperature (32 °C for one hour followed by the incubation at 52 °C for one hour) does not affect MTX fluorescence, confirming its thermostability (Figure 4c).

MTX efficiently targets and is well retained in the mitochondria of HEK cells and human fibroblasts (Figure S2). However, we did not notice any improvement over time when compared to MTY.

We then attempted to use MTX to measure mitochondrial temperature in human skin fibroblasts. The addition of oligomycin resulted in a sharp fluorescence decrease, while the addition of KCN induced a fluorescence increase with a rapid kinetics compatible with a membrane potential-dependent variation (Figure 5a). These results indicate that in human fibroblasts, MTX presents the same sensitivity to the fluctuations of membrane potential as MTY despite the added chloroacetyl motif suggesting the lack of binding of this motif within mitochondria of human fibroblasts. This unexpected behavior of the dye, which we have not studied further, precludes the use of MTX for temperature assessment in these cells.

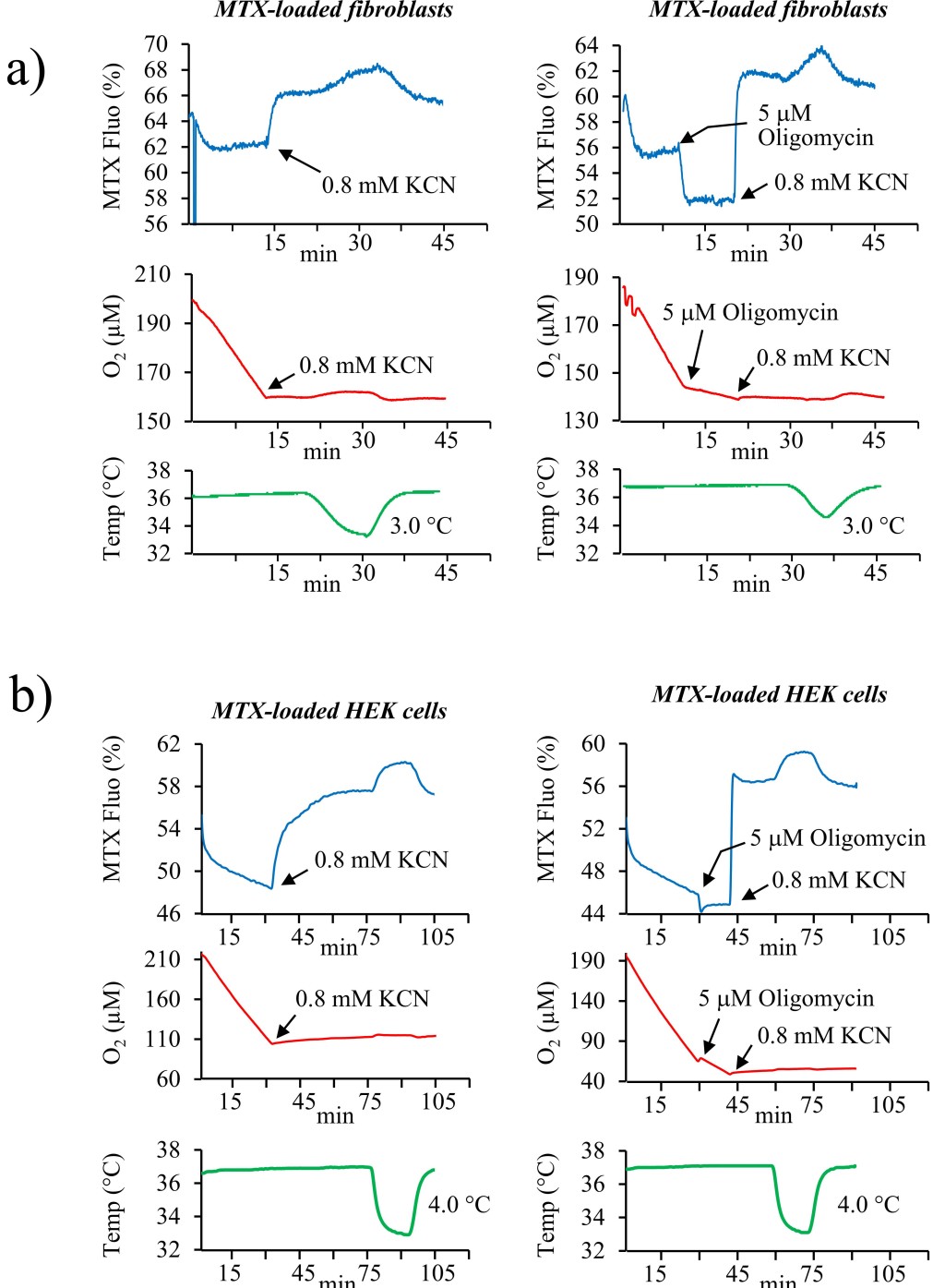

**Figure 5.** MTX and mitochondrial temperature in human skin primary fibroblasts and in HEK293 cells. Fluorescence changes in MTX-labeled human primary fibroblasts (**a**) and HEK293 cells. (**b**) Activation of the respiration causes a decrease in MTX fluorescence (blue lines). The inhibition of cell respiration by KCN results in a rapid increase in MTX fluorescence (blue line, left panel) or in a further decrease of the fluorescence upon oligomycin addition (blue line, right panel). Addition of KCN after oligomycin causes an immediate increase of the fluorescence. The respiration (red lines) and the temperature of the cell suspension medium (green lines) are simultaneously monitored. Measurements are carried out in a closed quartz chamber. Representative graphs from 4 (fibroblasts/KCN), 3 (fibroblasts/oligomycin), 2 (HEK/KCN), and 3 (HEK/oligomycin) independent measurements are shown.

Surprisingly, compared to MTY, the modification of the MTX structure which should have increased its binding and stability inside the mitochondria, in HEK cells, apparently altered the binding of the probe to its mitochondrial targets and made it sensitive to membrane potential variations (Figure 5b). Indeed, in MTX-labeled HEK cells, oligomycin addition resulted in a slight fluorescence decrease corresponding to MMP increase. The kinetics of the fluorescence change of MTX in HEK cells after KCN-induced respiration blockage was intermediate between the fast kinetics related to the membrane potential changes and the slow ones attributable to temperature changes. This probe behavior suggesting a response to both the temperature and the membrane potential and it might indicate that only a fraction of the probe is bound to its mitochondrial targets in the HEK cells. Nevertheless, as with human fibroblasts, MTX cannot be used to analyze mitochondrial temperature in HEK cells.

## 4. Discussion

As reported in our original work on HEK cells [8], and extended here to HeLa cells, the thermosensitive fluorescent MTY probe allows measuring the mitochondrial temperature in living cells. We have now shown in both cell types that actively respiring human mitochondria increase the temperature by about 10 °C. However as the temperature detected with the probe reflects its close surrounding, it might differ from that of the whole mitochondria. Therefore, the measured temperature should be considered locally, as that of the probe's surroundings, especially as its precise mitochondrial location is not known. Based on the charged nature of the dye, it most likely accumulates at the matrix side of the inner membrane, or within the inner membrane, bound to its targets, including ALDH2, one of known mitochondrial target of MTY [18]. In this respect, the binding of MTY might also depend on the environment provided by the mitochondria (as pH, density and ionic strength) and may vary according to the cell types and their metabolic status [23]. Thus, caution should be taken before extrapolating the measured temperature of 50 °C to the whole mitochondrial network and to the several intra mitochondrial compartments. Indeed, there might be spatial and temporal zones of elevated temperature in a similar manner to what was recently shown for cristae membrane potential and activity, which can vary independently within the same mitochondrion [28]. Being the source of heat, the operating respiratory chain within the inner membrane is likely maintained at elevated temperature. This is supported by the finding that the maximum activities of the respiratory chain complexes fall within the 50 °C range [8]. This was recently corroborated by the report of the high and unique stability of the respiratory chain proteins across species (Meltome atlas) [29]. The presence of mitochondrial cardiolipins involved in phase transition, the abundance of mitochondrial heat shock proteins possibly acting as thermostabilizers of protein structures, and the presence of thermo-protectant solutes (reviewed in [30]) further support the concept that mitochondria sustain and operate at elevated temperatures. Finally, using different biochemical approaches, not based on charged fluorophores, large temperature differences with surrounding cytosol were observed when uncoupling mitochondria from HeLa cells [6,31] or from aplasia neurons [32]. Nevertheless, until the unmasking of the exact mitochondrial distribution of these nanothermometers, caution is needed before considering a global mitochondrial temperature.

A second constraint related to thermosensors, such as MTY, is their dependence on the intrinsic binding sites within mitochondria for an efficient anchoring. Indeed, as positively charged molecules, they readily accumulate inside the mitochondria being attracted by the negative charges on the matrix side of the inner membrane. However, because mitochondrial potential fluctuates, reflecting dynamic changes in metabolism and bioenergetics status, charged dyes might relocalize according to these variations. Under conditions where membrane potential collapses, they may even diffuse out of mitochondria. Thus, only the ability of MTY to form stable bonds with proteins or other macromolecules allows its immobilization inside the mitochondria, making its location independent of membrane potential variations. As the efficient anchoring of the probe is a prerequisite for a local temperature monitoring, the presence of such binding sites is essential, yet variable according to the cell type.

While they appeared to be present in HEK and HeLa cells, allowing for MTY anchoring and temperature measurement, these targets seem to be absent or to have a low affinity for the dye in primary fibroblasts, which could explain the observed lability of MTY and its sensitivity to membrane potential variations in these cells. The same is true for MTX, a compound derived from MTY, to which a chloroacetyl motif was added to favor covalent binding with nearby proteins. However, this chemical modification did not ensure its stable binding into mitochondria, as the probe fluorescence readily follows the membrane potential variations. Moreover, this modification appears to change the affinity of the original MTY probe for its binding sites in the mitochondria of HEK cells, and consequently makes it sensitive to the variations of membrane potential as well. The dependence on these intrinsic binding sites and on their affinity for the thermosensitive probes renders the monitoring of the temperature impossible in these cells. Thus new tools are needed to obtain controlled and stable mitochondrial targeting in any cell type and different culture condition.

Finally, based on theoretical considerations, some concerns have been raised against the possibility of maintaining a temperature gradient between mitochondrial subcompartments, and the cytosol [33,34], discussed in [16,35]. However, their validity appears undermined by recently proposed models of mitochondrial thermogenesis as a special case of the thermal diffusion equation and by the experimental data demonstrating the occurrence of intracellular temperature differentials [6,8,31,32,36,37]. Finally, the internal mitochondrial membrane is mainly composed of highly packed proteins, with a quite low lipid content resulting in a distinct mitochondria-specific protein/lipid ratio [38]. The inner membrane is also organized into heterogeneous functional units, unequally distributed between the subfractions of the inner mitochondrial membrane, which we are only beginning to unravel. Thus, better understanding of the physicochemical properties of mitochondrial membranes together with development of new tools increasing the binding of intracellular nanothermometers should be the focus of future research.

## 5. Conclusions

Thermosensitive positively charged fluorophores such as MitoThermo Yellow (MTY) and its modified version MitoThermo X (MTX) have been developed for assessing the temperature of mitochondria in living cells. Their use as nanothermometers is dependent on their long term retention in mitochondria, lack of interference with mitochondrial activities, and on their propensity to be unaffected by fluctuations of membrane potential, which is likely enabled by their binding to specific targets within mitochondria. Using MTY, first in HEK cells and here in HeLa cells, we confirmed that operating mitochondria increase their temperature locally by about 10 °C and are maintained at close to 50 °C. However, the sensitivity of MTY to mitochondrial membrane potential variations in human primary fibroblasts and of MTX in both human primary fibroblasts and HEK cells, precluded their use for temperature determination in human fibroblasts and in both cell types, respectively. Although very easy to apply and highly responsive to the temperature variations, the use of these charged dyes as mitothermometers currently appears to be restricted to only certain cell types. Thus, caution is needed when evaluating mitochondrial temperature using these probes in different cell types and under various experimental conditions.

**Supplementary Materials:** The following are available online at http://www.mdpi.com/2227-9040/8/4/124/s1, Figure S1. Synthesis of MTX. Reagents and conditions. Spectral analysis of MTX. Figure S2: Mitochondrial staining of HEK293 cells and human primary fibroblasts with the thermosensitive MTY and MTX probes.

**Author Contributions:** Conceptualization, D.C., P.B., P.R., and M.R.; methodology, D.C., P.B., S.P., J.Y.L., Y.-T.C., P.R., and M.R.; validation, D.C., P.B., P.R., and M.R.; formal analysis, D.C., S.P., J.Y.L., Y.-T.C., P.R., and M.R.; investigation, D.C., P.B., P.R., and M.R.; resources, C.L., R.E.-K., S.P., J.Y.L., Y.-T.C., and G.L.; writing—original draft preparation, P.R. and M.R.; writing—review and editing, D.C., P.B., R.E.-K., S.P., J.Y.L., Y.-T.C., G.L., P.R., and M.R.; visualization, S.P., J.Y.L., Y.-T.C., D.C., P.R., and M.R.; supervision, P.R. and M.R. All authors have read and agreed to the published version of the manuscript.

**Funding:** This research received no specific external funding. General expenses were covered by grants from AAJI (Association pour l'Aide aux Jeunes Infirmes et aux Personnes Handicapées); AFAF (Association Française de

l'Ataxie de Friedreich); Association OLY (Ouvrir Les Yeux); ANR MITOXDRUGS (ANR-16-CE18-0010). Sponsors or funders had no role in the study design, data collection and analysis, decision to publish, or preparation of the manuscript.

**Conflicts of Interest:** The authors declare no conflict of interest.

## Abbreviations

HEK, Human Embryonic Kidney; KCN, Potassium Cyanide; MTG, MMP, Mitochondrial Membrane Potential; MitoTracker Green; MTR, MitoTracker Red; MTY, MitoThermo Yellow; MTX, MitoThermo X; TMRM, Tetramethylrhodamine, Methyl Ester, Perchlorate.

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
