# Peer review of "Pitfalls in Monitoring Mitochondrial Temperature Using Charged Thermosensitive Fluorophores"

_chemosensors, doi:10.3390/chemosensors8040124_

Round 1

Reviewer 1 Report

The authors describe the use of temperature sensitive fluorescence probes to characterize the environment around mitochondria. Mitochondria play an essential role in cellular function, through production of ATP, and their dysfunction is associated with several disease states, including diabetes and neurodegeneration. Therefore, understanding each aspect of their contribution to cellular homeostasis is paramount in fully understanding and developing therapeutics towards said disease states. The authors use fluorescence emission from multiple probes, MTX and MTY, which localize the mitochondria cell walls and have sensitized emission to their environment. The researcher find that the temperature around mitochondria in both HEK293 and HeLa cells is ~ 50C. Furthermore, they detail differences in responsiveness of probes depending upon cell type.  The overall manuscript is interesting with the fact that they clearly demonstrate that scrutiny needs to be present when using probes. However, there are several issues that need to be addressed before it can be considered for publication.

  • The resolution of the figures need to be adjusted. I could barely read the text in them and the lines were difficult to follow.
  • Line 46-48. “affecting thermal environment appears as a central mitochondrial function”. The authors provide no reference for this statement nor any data to verify the claim. Is this truly a know function of speculation? If known, please provide references.
  • Line 54: missing bracket (reviewed in [2].
  • 163-187: Figure 1 legend. This is just a repeat of what is stated in the results section. Modify the information and reduce the text
  • It doesn’t appear that the authors used any metabolic assays to verify the conditions of the mitochondria. This reviewer recommends that the authors have some control experiments, such as OCR, to establish viability of mitochondria and their changes in response to drug treatments.

Author Response

We thank the reviewers for her/his careful review of our manuscript and comments on our work. We modified our manuscript accordingly. We supply a new version of our manuscript as well as a marked version for review purposes where we highlighted in yellow the modifications to the text introduced in response to specific points raised by reviewers. All corrected figures are shown in both versions.

  1. The resolution of the figures need to be adjusted. I could barely read the text in them and the lines were difficult to follow.

All the figures are in the resolution of 820 or 1000 dpi (the recommended resolution being minimum 600 dpi) but indeed, in both word and pdf files, they appear in much lower resolution. We agree that it needs to be adjusted. As the figures are also provided as original tiff files of high resolution, they should be used in final manuscript, if accepted.

  1. Line 46-48. “affecting thermal environment appears as a central mitochondrial function”. The authors provide no reference for this statement nor any data to verify the claim. Is this truly a known function of speculation? If known, please provide references.

The local elevation in mitochondrial temperature shown in the present and previous study, which results from its thermogenic activity likely affects local thermal environment of mitochondria. However, new molecular tools are needed to precisely measure the temperature of intra-mitochondrial compartments and to verify this hypothesis. To avoid a confusion, we modified the statement. (Now: line 50-51)

  1. Line 54: missing bracket (reviewed in [2].

This has been corrected. (Now: line 56)

  1. 163-187: Figure 1 legend. This is just a repeat of what is stated in the results section. Modify the information and reduce the text.

The figure legend has been modified accordingly. (Now: line 168-187)

  1. It doesn’t appear that the authors used any metabolic assays to verify the conditions of the mitochondria. This reviewer recommends that the authors have some control experiments, such as OCR, to establish viability of mitochondria and their changes in response to drug treatments.

We agree that controlling the functionality of mitochondria in response to any treatment is crucial. For this reason we have developed an original set up allowing for simultaneous measurement of mitochondrial respiration to control mitochondrial coupling efficiency, sensibility to various mitochondrial inhibitors and the impact of any molecule on mitochondrial function in every assay. Thus, we could verify and confirm that MTY at concentration used in this work (and at least up to 300 nM) does not interfere with mitochondrial function, neither in intact nor in permeabilized cells. This statement is now included (197-202).

Reviewer 2 Report

The authors report on the “Pitfalls in monitoring mitochondrial temperature using charged thermosensitive fluorophores.” This is a continuation of their previously published work (ref 1) that showed that “mitochondria within human cells are maintained at close to 50 degrees C when active, increasing locally their temperature by about 10 degrees C.” Their previously published work used a positively charged rhodamine based fluorescent probe that they call MitoThermo Yellow (MTY). Their previously published work was performed using HEK293 cells. The authors note in the introduction that “At present, accurate real-time monitoring of mitochondrial temperature  in living cells remains quite challenging” and they cite a  relevant review of other temperature-dependent dyes (similar to their MTY), noting that these probes are “Among few molecular tools available” for real-time mitochondrial temperature measurements in living cells.

In this work the authors seek to extend the use of their MTY probe to other cell lines and did have partial success. In this work MTY was shown to reliably respond to mitochondrial temperature in HeLa cells. However, MTY did not reliably measure mitochondrial temperature in primary skin fibroblast. The authors attribute this to “the reactivity of MTY to membrane potential variations in human primary skin fibroblasts.”  A second positively charged rhodamine based fluorescent probe (a chemically modified version of MTY where “The chloroacetyl motif added to MTY to produce MTX can covalent bind to cysteine residues of nearby proteins…”) was investigated with the hopes that it would reliably measure mitochondrial temperature in a wider variety of cell types, ideally leading to a universal probe for measuring mitochondrial temperature in all cell lines. “MTX did not result in a temperature probe unresponsive to membrane potential variations that could be universally used in any cell type to determine mitochondrial temperature.” Section 3.3 “Modification of MTY into MTX did not generate a universal temperature probe incentive to membrane potential variations” described thoroughly these negative results. The paper concludes “Although very easy to apply and highly responsive to the temperature variations, the use of these charged dyes as mitothermometers appears to be restricted to only certain cell types. Thus cautions are needed when evaluating mitochondrial temperature using these probes in different cell types and under various experimental conditions.”

Some strong aspects of this paper include:

The authors are building on previously published results with worthy and realistic objectives.

The authors did demonstrate wider use of their probe (MTY) in an additional cell line.

Their conclusions were not overstated, and their negative results and caveats are important information that others in the field doing similar work should be aware of. Their work does “imply cautiousness while using these nanothermometers for mitochondrial temperature analysis.”

Some drawbacks and points that should be addressed include:

The structures of these probes should be included in the main document rather than in the supporting information.

A section outlining the origins and/or preparation and characterization of any non-commercially available probes should be included in the Materials and Methods section. Citing a previous preparation would be sufficient. One assumes the preparation, isolation, and characterization of MTY was reported in their previously published work (if it was a new compound). If not, this must be reported. If it was not a new compound, the origins of the material used in this work should be reported. Importantly, the authors seem to imply that MTX is a new compound and no characterization was included. I cannot recommend publication of a paper including results using a new compound without adequate documentation of is preparation and characterization. If MTX is not a new compound, this should be clearly stated, and the origin of the material used in this work should be reported. Results and conclusions remain in question unless it can be demonstrated that the materials used match the structures reported.

It appears the abbreviation for “mitochondrial membrane potential”, MMP is defined in figure 1 caption rather than in the text where MMP is also used. The document would be more readable if abrasions are defined in the text.

Author Response

We thank the reviewers for her/his careful review of our manuscript and comments on our work. We modified our manuscript accordingly. We supply a new version of our manuscript as well as a marked version for review purposes where we highlighted in yellow the modifications to the text introduced in response to specific points raised by reviewers. All corrected figures are shown in both versions.

  1. The structures of these probes should be included in the main document rather than in the supporting information.

This is now included in the main document.

  1. A section outlining the origins and/or preparation and characterization of any non-commercially available probes should be included in the Materials and Methods section. Citing a previous preparation would be sufficient. One assumes the preparation, isolation, and characterization of MTY was reported in their previously published work (if it was a new compound). If not, this must be reported. If it was not a new compound, the origins of the material used in this work should be reported. Importantly, the authors seem to imply that MTX is a new compound and no characterization was included. I cannot recommend publication of a paper including results using a new compound without adequate documentation of is preparation and characterization. If MTX is not a new compound, this should be clearly stated, and the origin of the material used in this work should be reported. Results and conclusions remain in question unless it can be demonstrated that the materials used match the structures reported.

MTY is indeed a previously reported probe (Arai et al. 2015) and its origin is now clearly cited in the Material and Methods section.

On the other hand, MTX is a new compound and its synthesis and characterization is now presented in the supplemental data sections.

  1. It appears the abbreviation for “mitochondrial membrane potential”, MMP is defined in figure 1 caption rather than in the text where MMP is also used. The document would be more readable if abrasions are defined in the text.

This is now corrected and MMP abbreviation is defined in the main text as it appears for the first time (161) as well as in the Abbreviation section.

Reviewer 3 Report

In this manuscript the authors evaluate fluorescent dyes as probes to monitor mitochondrial temperature. Building on initial results with MitoThermo Yellow (MTY) in HEK293 cells, similar experiments are conducted with HeLa cells and similar results are obtained. MTY was not able assess mitochondrial temperature in human primary skin fibroblasts. The authors suggested introducing a group to covalently attach the probe to the membrane should increase the stability of the probe and its retention in the mitochondria. The authors determined that MTX was sensitive to membrane potential variations in HEK cells and showed complicated behavior in fibroblasts.

While this manuscript may be suitable for publication, several issues first need to be addressed. A primary concern is the negative result for MTX as a probe. The authors comment on the chloroacetyl group to allow the compound to covalently interact with the member, but there is no evidence to support that this actually occurs. The complications with MTX in HEK cells suggest that this probe may be problematic. Without a ‘positive’ result in HEK or HeLa cells it is difficult to establish that MTX may be useful as a probe. This finding fits into the pitfalls as noted by the authors in the title, but this should be more clearly stated -the observations for the new compound tested was complicated for both HEK cells and fibroblasts.

It is not clear from the manuscript if MTX is commercially available (and was commercially obtained), or if the compound was synthesized as part of this project. Invitrogen catalog numbers were provided for MTG and MTR, but unless missed not for MTY and MTX. If MTX was synthesized as part of this project, then characterization (NMR and mass spec) are needed.

Much of the beginning of the manuscript describes the use of MTY in HEK cells, and it was noted that this result was presented in a previous manuscript. The authors need to clearly delineate what the advancement in this manuscript is regarding MTY use in HEK cells.

A challenge when comparing results in the figures is that the y-axes are often different for different plots (such as 37-53% versus 37-49% in Figure 1a). While it is understandable to have the axes scaled for clarity in the size of the figure, using different axes make it difficult to qualitatively and quantitatively compare the different conditions.

Minor points:

-Line 94 – Invitrogen number is M7512

-Line 184 indicates a massive difference. It is not clear what defines a difference as massive

-Figure 2 referenced in the manuscript before Figure 1

-MTX can be shown in the cationic form to be consistent with the other structures presented

-Figure S2 MTY – to be consistent with bond line structures and the other structures the terminal methyl groups should not be shown as ‘CH3

There are numerous grammatical throughout the manuscript. A few examples are:

Line 39 – delete ‘up’

Line 53 ‘Among the few molecular…’

Line 54 – missing closing of parentheses

Line 57 – conditions

Line 120 – missing comma to end appositive

Line 136 – subject/verb agreement – should read ‘are’

Line 196 – ‘to control for the effect of the inhibitors..’

Line 264 – should read ‘in the mitochondria’

Line 276 – should read Figure S1

Line 345 – should read binding

Author Response

We thank the reviewers for her/his careful review of our manuscript and comments on our work. We modified our manuscript accordingly. We supply a new version of our manuscript as well as a marked version for review purposes where we highlighted in yellow the modifications to the text introduced in response to specific points raised by reviewers. All corrected figures are shown in both versions.

  1. A primary concern is the negative result for MTX as a probe. The authors comment on the chloroacetyl group to allow the compound to covalently interact with the member, but there is no evidence to support that this actually occurs. The complications with MTX in HEK cells suggest that this probe may be problematic. Without a ‘positive’ result in HEK or HeLa cells it is difficult to establish that MTX may be useful as a probe. This finding fits into the pitfalls as noted by the authors in the title, but this should be more clearly stated -the observations for the new compound tested was complicated for both HEK cells and fibroblasts.

As pointed by the reviewer, MTX cannot be used to assess mitochondrial temperature in neither human fibroblasts nor HEK cells and it indeed falls into pitfalls in mitochondrial temperature measurements reported in this work. We now stated it more clearly modifying the text accordingly (273, 291-296, 392-394).

  1. It is not clear from the manuscript if MTX is commercially available (and was commercially obtained), or if the compound was synthesized as part of this project. Invitrogen catalog numbers were provided for MTG and MTR, but unless missed not for MTY and MTX. If MTX was synthesized as part of this project, then characterization (NMR and mass spec) are needed.

As noticed by the reviewer, neither MTY nor MTX are commercially available.

MTY is a compound previously reported in Arai et al. 2015. This original work is now cited in the Material and Methods section.

MTX is a new compound and its synthesis and characterization is now presented in the supplemental data sections.

  1. Much of the beginning of the manuscript describes the use of MTY in HEK cells, and it was noted that this result was presented in a previous manuscript. The authors need to clearly delineate what the advancement in this manuscript is regarding MTY use in HEK cells.

In our first study we used HEK cells in order to set up experimental conditions allowing mitochondrial temperature analysis with MTY. In the current study we have included a set of novel experiments validating the use of MTY for temperature measuring in these cells. Indeed, by using various mitochondrial membrane potential (MMP)-specific probes such as MTR and TMRM, we attempted to show that MMP variation can be consistently detected in both, intact and permeabilized HEK cells. Thus, the distinct behaviour of MTY not responding to MMP further validates its use for temperature monitoring. It is now stated more clearly (190-194).

  1. A challenge when comparing results in the figures is that the y-axes are often different for different plots (such as 37-53% versus 37-49% in Figure 1a). While it is understandable to have the axes scaled for clarity in the size of the figure, using different axes make it difficult to qualitatively and quantitatively compare the different conditions.

We agree with the reviewer that ideally the axes of the plots should be identical. Thus, for most figures, the graphs to be compared show similar axes (% of fluorescence or oxygen concentration). However, for some plots as pointed by reviewer the axes have been re-scaled for clarity. The actual quantification of these fluorescent signals is based on the calibration with waterbath-controlled cell suspension medium temperature at the end of each experiment as shown in all graphs. This should allow the reader to compare the results obtained under different conditions. However, taking into consideration the reviewer comment, the graphs have been modified wherever it did not obscure the reading.

Minor points:

  1. Line 94 – Invitrogen number is M7512

It must have been some misunderstanding as Invitrogen reference for MitoTracker Green is M7514 as indicated. M7512 is Invitrogen number for MitoTracker Red that we are using as well and which is also referenced in the Material and Methods section.

  1. Line 184 indicates a massive difference. It is not clear what defines a difference as massive

We agree with the reviewer that “massive” may not be an appropriate way to describe fluorescence changes. This has been removed.

  1. Figure 2 referenced in the manuscript before Figure 1

This has been corrected and figures are now referenced in the right order.

  1. MTX can be shown in the cationic form to be consistent with the other structures presented
  2. Figure S2 MTY – to be consistent with bond line structures and the other structures the terminal methyl groups should not be shown as ‘CH3

 Both figures have been modified accordingly and figure S2 has been fused into Figure 4 as requested by a reviewer.

  1. There are numerous grammatical throughout the manuscript. A few examples are:

 Line 39 – delete ‘up’

 Line 53 ‘Among the few molecular…’

 Line 54 – missing closing of parentheses

 Line 57 – conditions

 Line 120 – missing comma to end appositive

 Line 136 – subject/verb agreement – should read ‘are’

 Line 196 – ‘to control for the effect of the inhibitors..’

 Line 264 – should read ‘in the mitochondria’

 Line 276 – should read Figure S1

 Line 345 – should read binding

The manuscript was revisited and corrected.

Round 2

Reviewer 2 Report

The authors have now included the structures of the probes used including thier previously reported MTY, their newly reported MTX, and commercially available MTR in the main text of the manuscript as this reviewer requested. They have also included in Supplemental material the minimum characterization of their newly reported compound, MTX, as this and another reviewer requested. There were also some questions about what was new as related to MTY and about the source of MTY in this work. The authors added a reference to their previous work on MTY in the Materials and Methods section, although without any explanation why the reference was added (line 84). A statement such as “MTY was prepared as previously reported [18]” would help to clarify to the reader that MTY is not commercially available and was previously reported by these authors.  Interestingly, this reference to MTY and newly included material related to the origin of MTX “(synthetic procedure and chemical analysis of MTX in Supplemental materials)” seems oddly buried within “The Staining procedures and life cell imaging” section (line 84-85) within Material and Methods. On a side note, is there a typo referring to “life” instead of “live” in this section heading (line 79)? This reviewer would have expected that the sourcing and/or preparation of the probes might have merited a brief section of their own in the Material and Methods section to make the information more easy to locate.

Additionally, it is this reviewer’s opinion that the authors may have made only a cursory effort to address some other reviewers’ comments and concerns. For instance, one of the other reviews notes “A primary concern is the negative result for MTX as a probe. The authors comment on the chloroacetyl group to allow the compound to covalently interact with the member, but there is no evidence to support that this actually occurs. The complications with MTX in HEK cells suggest that this probe may be problematic. Without a ‘positive’ result in HEK or HeLa cells it is difficult to establish that MTX may be useful as a probe. This finding fits into the pitfalls as noted by the authors in the title, but this should be more clearly stated -the observations for the new compound tested was complicated for both HEK cells and fibroblasts."  The authors added the words “is expected” on line 273, and “attempted to use” on line 291, and slightly modified a sentence in their conclusion (lines 392-394) to address this set of concerns.  There are ample experiments that could have been performed to demonstrate that their new probe containing a chloroacetyl group, MTX, either does or does not covalently interact with cysteines or cysteine residues. In the absence of actually performing these experiments, one might reasonably expect that literature precedent for these interactions (or lack thereof) would be included.

Author Response

We thank the reviewers for her/his comments on our work. We have answered the additional questions and modified our manuscript accordingly.

1) The authors have now included the structures of the probes used including thier previously reported MTY, their newly reported MTX, and commercially available MTR in the main text of the manuscript as this reviewer requested. They have also included in Supplemental material the minimum characterization of their newly reported compound, MTX, as this and another reviewer requested. There were also some questions about what was new as related to MTY and about the source of MTY in this work.

The authors added a reference to their previous work on MTY in the Materials and Methods section, although without any explanation why the reference was added (line 84). A statement such as “MTY was prepared as previously reported [18]” would help to clarify to the reader that MTY is not commercially available and was previously reported by these authors.

       The suggested statement and 2 references are now added to clarify that point.

2) Interestingly, this reference to MTY and newly included material related to the origin of MTX “(synthetic procedure and chemical analysis of MTX in Supplemental materials)” seems oddly buried within “The Staining procedures and life cell imaging” section (line 84-85) within Material and Methods. On a side note, is there a typo referring to “life” instead of “live” in this section heading (line 79)? This reviewer would have expected that the sourcing and/or preparation of the probes might have merited a brief section of their own in the Material and Methods section to make the information more easy to locate.

     We did not intend to undermine the importance of the synthesis and chemical analysis of the new probe, MTX. As suggested, we have now devoted to it a section in the Material and Methods.

3) Additionally, it is this reviewer’s opinion that the authors may have made only a cursory effort to address some other reviewers’ comments and concerns. For instance, one of the other reviews notes “A primary concern is the negative result for MTX as a probe. The authors comment on the chloroacetyl group to allow the compound to covalently interact with the member, but there is no evidence to support that this actually occurs. The complications with MTX in HEK cells suggest that this probe may be problematic. Without a ‘positive’ result in HEK or HeLa cells it is difficult to establish that MTX may be useful as a probe. This finding fits into the pitfalls as noted by the authors in the title, but this should be more clearly stated -the observations for the new compound tested was complicated for both HEK cells and fibroblasts."  The authors added the words “is expected” on line 273, and “attempted to use” on line 291, and slightly modified a sentence in their conclusion (lines 392-394) to address this set of concerns.  There are ample experiments that could have been performed to demonstrate that their new probe containing a chloroacetyl group, MTX, either does or does not covalently interact with cysteines or cysteine residues. In the absence of actually performing these experiments, one might reasonably expect that literature precedent for these interactions (or lack thereof) would be included.

     The chloroacetyl motif is known to react with thiols in the cell (we have now included a reference: Lee et al. 2018) and we expected that it would covalently bind to the cysteines of nearby proteins thus increasing the dye’s retention in the mitochondria. It seems not to be the case in the mitochondrial environment of human fibroblasts or in HEK cells. We stated it more clearly (298-302, 320-326, 391-397).

Reviewer 3 Report

The authors addressed the concerns noted in the initial review.

Author Response

English language and style are fine/minor spell check required

The manuscript was revisited and corrected.